# The Impact of Harsh Stratospheric Conditions on Survival and Antibiotic Resistance Profile of Non-Spore Forming Multidrug Resistant Human Pathogenic Bacteria Causing Hospital-Associated Infections

**DOI:** 10.3390/ijerph20042787

**Published:** 2023-02-04

**Authors:** Ignacy Górecki, Agata Kołodziejczyk, Matt Harasymczuk, Grażyna Młynarczyk, Ksenia Szymanek-Majchrzak

**Affiliations:** 1Department of Medical Microbiology, Medical University of Warsaw, Chalubinskiego, Str. 5, 02-004 Warsaw, Poland; 2Analog Astronaut Training Center, Morelowa Str. 1F/4, 30-222 Cracow, Poland; 3Space Technology Centre, AGH University of Technology, Czarnowiejska Str. 36, 30-054 Cracow, Poland

**Keywords:** antibiotic resistance, astrobiology, hospital-associated infections (HAI), human pathogenic bacteria, stratospheric flight

## Abstract

Bacteria are constantly being lifted to the stratosphere due to air movements caused by weather phenomena, volcanic eruptions, or human activity. In the upper parts of the atmosphere, they are exposed to extremely harsh and mutagenic conditions such as UV and space radiation or ozone. Most bacteria cannot withstand that stress, but for a fraction of them, it can act as a trigger for selective pressure and rapid evolution. We assessed the impact of stratospheric conditions on the survival and antibiotic resistance profile of common non-spore-forming human pathogenic bacteria, both sensitive and extremely dangerous multidrug-resistant variants, with plasmid-mediated mechanisms of resistance. Pseudomonas aeruginosa did not survive the exposure. In the case of strains that were recovered alive, the survival was extremely low: From 0.00001% of *Klebsiella pneumoniae* carrying the *ndm*-1 gene and methicillin-resistant *Staphylococcus aureus mec*A-positive with reduced susceptibility to vancomycin (MRSA/VISA), to a maximum of 0.001% of *K. pneumoniae* sensitive to all common antibiotics and *S. aureus* sensitive to vancomycin (MRSA/VSSA). We noticed a tendency towards increased antibiotic susceptibility after the stratospheric flight. Antimicrobial resistance is a current real, global, and increasing problem, and our results can inform current understandings of antibiotic resistance mechanisms and development in bacteria.

## 1. Introduction

Hospital-acquired infections (HAI) are infections that occur during patients’ hospitalization. They are noticed most frequently in intensive care units (ICU); however, every hospital environment is prone to HAI spreading [1]. They cause serious complications and are becoming a serious issue, especially in developing countries. Nowadays, in the age of increased antibiotic resistance, empiric, frequent, multidrug, prolonged, and/or inadequate therapies favor the selection and spreading of resistant bacteria in hospital environments and cause unsuccessful treatment [1,2,3]. HAI has been suggested to contribute to increased mortality and prolonged hospitalization [4,5].

The Centers for Disease Control and Prevention in the USA have listed several bacteria species that cause the most dangerous HAIs. *Staphylococcus aureus* is a Gram-positive, coagulase-positive coccus. It is commonly found as a harmless commensal, but it is also the second leading cause of HAI in the USA [6]. Methicillin-resistant *S. aureus* (MRSA) is especially associated with a worse prognosis. Its resistance is provided by *mec*A, *mec*C, or *mec*B genes located on SCC*mec* (Staphylococcal chromosome cassettes *mec*) [7,8,9]. The *mec*A gene encodes a modified penicillin-binding protein (PBP2a), with low affinity to almost all members of that antibiotic group [10,11]. Vancomycin is the ‘drug of choice’ to treat MRSA infections, and there is much concern regarding the emergence of *S. aureus* strains that are resistant to vancomycin. The intermediate-resistant *S. aureus* (VISA) phenotype is associated with increased cell wall thickness and changes in the expression of several genes. In 2002, a greater threat was described: Vancomycin-resistant *S. aureus* (VRSA), which is a high-level resistant variant containing an enterococcal *van*A operon. Bacteria carrying this operon synthesized modified D-Ala-D-lactate terminated peptidoglycan precursors with low affinity to vancomycin making them resistant to vancomycin [12].

*Klebsiella pneumoniae* and *Escherichia coli* are Gram-negative rods classified in the *Enterobacteriaceae* family. *K. pneumoniae* is found primarily in hospital environments, whereas *E. coli* is a common microorganism present in the outside environment as well as in the human gastrointestinal tract. Although *E. coli* is a commensal, both organisms can cause infections such as urinary tract and bloodstream infections, as well as others. *K. pneumoniae* and *E. coli* are the third and fourth leading causes of HAI in the USA, respectively [6]. *Enterobacteriaceae* show a wide range of antibiotic resistance, both intrinsic and acquired. A particular challenge is their resistance to carbapenems. Several mechanisms play an important role in this antibiotype, but the most important is the expression of carbapenemases, such as *Klebsiella pneumoniae* carbapenemases (KPC) or metallo beta-lactamases (MBL). Among them, the New Delhi carbapenemase-producing strains are of particular importance as the most harmful and extreme drug-resistant (XDR) pathogens. Together with the upregulation of efflux pumps they can exhibit resistance to almost all antibiotics [13,14].

*Enterococcus faecalis* is a Gram-positive coccus. *Enterococcus* spp. colonize the human gastrointestinal tract but can also spread beyond this. *E. faecalis* is the fifth leading cause of HAI in the USA [6]. It is characterized by intrinsic resistance to and tolerance of many drugs. It can also survive on various surfaces and medical devices for long periods, which makes it dangerous in a hospital environment. Resistance to glycopeptides is the most concerning feature. It is determined by a few gene clusters (*van*A—*van*M). Vancomycin-resistant *Enterococci* (VRE) with the *van*A operon (the same operon that mediates resistance in VRSA) show complete high-level resistance both to vancomycin and teicoplanin, whereas bacteria with *van*B operon are resistant only to vancomycin [15].

*Pseudomonas aeruginosa* is a Gram-negative bacilli. Its ability to adapt to various environments (e.g., by forming biofilms) makes it a dangerous pathogen in hospital surroundings. It is the sixth leading cause of HAI in the USA and is especially common in ventilation-associated pneumonia [6]. Several features of *P. aeruginosa* make it naturally resistant to antibiotics such as beta-lactams, aminoglycosides, and fluoroquinolones. Its ability to form biofilms, as well as its resistant cell membrane, developed efflux pump system, and lack of specific porins (OprD), makes it difficult for most antibiotics to penetrate the bacteria. It can also produce enzymes capable of inactivating antibiotics such as beta-lactamases or aminoglycoside-modifying enzymes (AMEs). On several occasions, it has been found that these mechanisms can be enhanced by mutations, resulting in increased resistance [16].

Naturally, the concentration of microorganisms decreases with altitude due to harsher conditions, but several stratospheric balloon and rocket missions showed that bacteria such as *Bacillus* spp. can be found in the upper layers of the atmosphere [17]. Most of these microorganisms are concentrated in the troposphere (up to 18 km above sea level, ASL), where conditions are somewhat hospitable. Bacteria are not only freely transported with wind currents while being metabolically active, but also were found in the nuclei of clouds being formed [18]. It is estimated that 10^21^ bacterial cells are lifted to the upper atmosphere every year due to weather phenomena such as storms, volcano eruptions, and human activity [17]. Conditions in the stratosphere (18–50 km ASL) are drastically different. Bacteria that are lifted to the stratosphere experience large-amplitude temperature variations. Throughout the troposphere, temperatures drop down to approximately −60 °C at the tropopause (the border between troposphere and stratosphere). Temperatures rise again up to approximately 0 °C at the edge of the mesosphere (ca. 50–85 km ABL). Atmospheric pressure and humidity gradually decrease with altitude to reach 0 within the stratosphere. At the upper border of the troposphere, bacteria are exposed to high concentrations of O_3_ molecules in the ozone layer. After crossing that layer, organisms are again exposed to yet another mutagenic factor—UVC (220–280 nm), which is normally blocked by the ozone layer. Moreover, space (ionizing) radiation increases with altitude. Only a fraction of organisms can withstand this environment [17]. However, it is worth noticing that such conditions favor stronger organisms and can accelerate selective pressure causing rapid evolution [18]. Structural changes in the genome, dysfunction in cellular proteins, and global regulatory system disorders acquired after vertical migration in the atmosphere together with migration of those organisms across the globe pose a possible issue for public health and infectious disease control [17,19]. Our study aimed to assess the impact of harsh stratospheric conditions on the survival and antibiotic resistance profile of common non-sporulating human pathogenic bacteria, both sensitive and extremely dangerous multidrug-resistant variants. Nowadays, antimicrobial resistance is a real, global, and constantly increasing problem. Our results can change the current common view of bacterial antibiotic resistance mechanisms and development.

## 2. Materials and Methods

### 2.1. Bacterial Cultures

Model bacterial strains were acquired from the ATCC collection: *Escherichia coli* (ATCC 25922), *Klebsiella pneumoniae* (ATCC 700603), *Pseudomonas aeruginosa* (ATCC 27853), *Enterococcus faecalis* (ATCC 29212), and *Staphylococcus aureus* (ATCC 25923), which were generally sensitive to antibiotics as well as clinical isolates (resistant variants of previously introduced species), with well-assigned mechanisms of resistance. These included carbapenem-resistant *E. coli* (CRE), carrying the *kpc* gene; carbapenem-resistant *K. pneumoniae* (CRKP) (*ndm*-1 positive); multidrug-resistant (MDR) *P. aeruginosa* with the *vim* gene; *E. faecium* (VRE) with the *van*A operon (henceforth referred to as ‘resistant’ variants), and *S. aureus* (MRSA/VSSA) and *S. aureus* (MRSA/VISA), *mec*A-positive. All isolates were obtained from stocks archived at the Department of Medical Microbiology, Medical University of Warsaw.

### 2.2. Sample Preparation and Pre-Flight Analyses

Experimental Design

Three sets of samples were prepared: The experimental, flight control, and ground control groups. Experimental and flight control groups were both placed in the payload; however, the flight control samples were covered with aluminum foil, which acted as protection from direct sunlight, UV, and ionizing radiation and provided thermal isolation [20]. The ground control group was kept on the ground in normal atmospheric conditions during the flight.

Bacterial Culture

From the log phase of growth of bacterial cultures of each strain, solutions with OD = 2.0 MacF (solution A) and OD = 0.5 MacF (solution B) were prepared. Samples were composed of 1 mL of solution A in 1.8 mL cryotubes. Each sample was centrifuged, the medium was removed, and the pellets were frozen at −70 °C.

Viable Count Assay

A viable count assay was performed with the use of 100 μL of solution A to prepare a series of dilutions. Bacteria from dilutions 10^−5^, 10^−6^, and 10^−7^ were cultured on agar plates and used to calculate Colony Forming Units (CFUs).

Antibiotic Resistance Determination

The determination of antibiotic resistance was performed using 1 mL of solution B for antibiogram preparation. Antibiotics were chosen according to The European Committee on Antimicrobial Susceptibility Testing (EUCAST) recommendations (antibiotics are listed in Appendix A). Antibiograms were performed using antibiotic diffusion discs (Oxoid, Basingstoke, UK) and E-tests (BioMérieux, Craponne, France).

### 2.3. Post–Flight Analyses

Bacterial Culture

When post-flight samples were recovered, they were transferred to the laboratory in −20 °C temperature conditions. To each sample, 1 mL of fresh, sterile Brain Heart Infusion (BHI) broth medium was added to create the suspension of the bacterial cells.

Viable Count Assay

From each sample, 100 μL of non-diluted suspension was cultured on an agar plate and 100 μL was used to prepare a series of dilutions. Dilutions of 10^−1^, 10^−2^, and 10^−3^ were cultured on agar plates and used to calculate CFU.

Antibiotic Resistance Determination

The remaining volume of the broth medium BHI suspension was incubated at 37 °C for 2 h before determining antibiotic resistance. One milliliter of that suspension was used according to the procedure that was performed pre-flight. Colonies that showed increased resistance (were present within the area of growth inhibition) or a different morphology were isolated. In total, 116 colonies were isolated (*E. coli*—7; CRE—10; *K. pneumoniae*—15; CRKP—14; *P. aeruginosa*—6; MDR *P. aeruginosa*—4; *E. faecalis*—11; VRE—9; *S. aureus*—6; MRSA/VSSA—22; MRSA/VISA—12).

PCR Analyses

Total genomic DNA was isolated from clinical isolates of resistant bacterial strains and isolates of resistant strains were collected after the flight using the Genomic DNA Purification Kit (A&A Biotechnology, Gdańsk, Poland) according to the manufacturer’s instructions. Gene-specific primers for genes (*kpc*, *ndm-*1, *vim van*B, and *mec*A) were used (Appendix A). The PCR was carried out using 5 μL of buffer for polymerase, 5 μL of MgCl2, 2 μL of the primer mix, 2 μL of deoxyribose nucleotide triphosphate (dNTP), 0.4 μL of polymerase, 2 μL of the DNA template, and 33.6 μL of deionized water. The total volume of each PCR mixture was 50 μL. The DNA amplification was performed using a Thermocycler C-1000 (Bio-Rad, Hercules, CA, USA) (for protocols see Appendix A). PCR products were loaded to a 1.5% agarose gel with ethidium bromide. The agarose gel electrophoresis was performed using a TAE buffer for 45 min at 120 V. The amplified DNA fragments were visualized using the Bio-Rad Molecular Imager GelDoc XR+ and analyzer Image Lab Software v4.0.1.

### 2.4. Stratospheric Flight (Technical Details)

The STRATOS mission was 2 h 29 min long and it was launched on 18 July 2020 at 10:30 CEST at the Queen Jadwiga Astronomical Observatory in Poland, with the following geographical coordinates: 49.7761 latitude and 21.0901 longitude. 4 m^3^ of hydrogen gas (Linde) was inflated into the 1600 g latex balloon. A scientific payload weighing 2 kg (with two cameras onboard and one tracker and parachute) was attached to the balloon. The balloon flight occurred according to the planned time because the flight predictions were correct in that the balloon landing was planned to be in a safe area within the borders of Poland. Flight predictions were made using predict.habhub.org software integrated with Google Maps, available online. The flight prediction pathway very accurately covered the actual balloon track, which is visualized in Appendix A. The balloon ascent time was 1 h 43 min, while the descent was only 46 min. The average ascent velocity was 4.825 m/s. The balloon burst at a 31 km altitude. The tracking system was provided by the SPOT GEN3 satellite GPS messenger. This device was used for flight tracking using 100% satellite technology. Unfortunately, the operating altitude for this device is 6500 m with a 1.6 GHz frequency, so we were unable to use it for measuring altitude. It was chosen because it is very accurate in locating the landed cargo on the ground surface. The altitude was measured based on other parameters including two independently recorded videos of the flight. Having the time of the recording, the time of the balloon burst (it was visible on the camera), the time of the balloon launch, and the time of the balloon landing and passing through the cloud zones, we could compute the velocities of the payload and based on them we computed the altitude. Additionally, we used a handmade electronic device to record data logs with a frequency of 1 s. This device was made by the Polish Rocket Society for stratospheric missions, and it contained 4 standard temperature sensors: Linear thermistors MCP9700-E/TO for Arduino.

Figure 1 depicts the STRATOS mission flight profile. The flight was 2.5 h long, with approximately 1 h exposure to high levels of UVA, UVB, and UVC light. The balloon burst occurred in the stratosphere at approximately 31 km altitude. The graph on the right side visualizes temperature fluctuations. Table 1 presents data logs collected at critical time points of the STRATOS mission flight revealing temperature extremes.

We used two types of UV sensors: ML8511, which detects 280–390 nm light the most effectively. The second sensor used was GUVA-S12SD 240–370 nm. Sensors were integrated with the Arduino board. Obtained data revealed changes of intensity in UV radiation dependent on the altitude (Figure 2). The ground level of UV sensor activity was 54 units in the Sun and 2 units in shadow. In the stratosphere, the maximum UV sensor activity was 396 in sunlight and 189 in shadow. After 8 min of the flight, UV sensor activity in sunlight was already above 250 units. Because the landing was in a cloudy area, UV activity unit drop was observed faster than 8 min before landing (32 min before landing). The total time of increased UV light exposure (above 250 units in the sunlight) was 1 h 49 min.

Another difficulty was that the capsule with biological samples was continuously rotating similar to a carousel, so UV light exposure was not constant, and we had to average the exposition time for higher levels of UV light by comparing maximal and minimal values and calculating the exposure ratio. The ratio of UV in the sunlight to UV in the shadow on the ground was 52/2 = 26. The same parameter for the stratosphere was 396/189 = 2.1. For the upper part of the atmosphere, it was 283/98 = 2.9. The highest values of UV light intensity occurred during 20 min of the flight (in this period, maximum values were above 375 UV sensor activity units in the sunlight), correlated with relatively high-temperature values. Assuming that UV radiation in the stratosphere has the maximum possible value (no clouds, no dust, and no gases), meaning 100%, 13% of this light spectrum reaches the Earth’s surface. During the flight, biological samples were exposed to a minimum of 63% of the UV light intensity for 1 h 49 min. This means that the average UV light exposure during the whole STRATOS flight was 55%, so more than 4 times larger than on the Earth. The ML8511 sensor did not reveal such altitude-related UV intensity differences, indicating that UVB penetrates the atmosphere much easier than shorter bands of the solar spectrum.

Passive dosimeters measuring ionizing radiation provided by the National Center for Nuclear Research in Cracow did not reveal increased doses of radiation during the flight (which we associate with a short time of exposure and limited sensitivity of the sensors).

## 3. Results

### 3.1. Viable Count Assay

Referring to the main objective of this study, the viability of bacterial strains before and after exposure to the stratospheric environment was assessed and compared. The results showed some tendency in the pattern of viability between bacterial strains. Both sets (the experimental group and flight control group) that were sent into the stratosphere had noticeably lower absolute CFU values compared to before the flight and ground control group; however, the viability of bacteria covered with aluminum foil (the flight control group) had better survival than bacteria without protection (Figure 3). Compared to the ground control group, the rate of survival of the flight control samples was 1.06–24.3 times lower (*E. coli*—8.93; *E. coli* (resistant)—1.06; *K. pneumoniae*—5.93; *K. pneumoniae* (resistant)—23.7; *P. aeruginosa*—5.16; *P. aeruginosa* (resistant)—3.64; *E. faecalis*—15.9; *E. faecalis* (resistant)—11.7; *S. aureus*—24.3; *S. aureus* (MRSA/VSSA)—23.7; *S. aureus* (MRSA/VISA)—6.21). In the experimental group, the susceptible variant of *E. coli* and both strains (susceptible and resistant) of *P. aeruginosa* did not survive the flight. In the same group, the survivability of other strains ranged from 92.95 to 1.23 × 10^6^ times lower than in the ground control group (*E. coli* (resistant)—92.95; *K. pneumoniae*—5.19 × 10^4^; *K. pneumoniae* (resistant)—7.9 × 10^5^; *E. faecalis*—1.63 × 10^5^; *E. faecalis* (resistant)—1.4 × 10^5^; *S. aureus*—1.1 × 10^5^; *S. aureus* (MRSA/VSSA)—1.1 × 10^4^; *S. aureus* (MRSA/VISA)—1.23 × 10^6^). Although most of the samples contained live bacteria after the flight, it is important to be aware that this was only a small fraction of the total. Table 2 shows the percentages (relative to initial inoculum) of alive bacteria after the flight.

### 3.2. Antibiotic Susceptibility

The second aim of our study was to examine if exposure to stratospheric conditions caused changes in the antibiotic resistance profiles of tested strains. A set of 32 antibiotics were used for the tests (a total list of drugs is shown in Appendix A). Proper sets of antibiotics (according to EUCAST guidelines) were used to create antibiograms for each tested species. Specifically, they were as follows: (i) For *E. coli* and *K. pneumoniae,* 15 and 14 antibiotics; (ii) for *P. aeruginosa,* 11; (iii) for *E. faecalis,* 10; and (iv) for *S. aureus,* 18 dedicated antibiotics. Clinical interpretation changed only in some cases (Appendix A), but we did observe changes in the diameters of growth inhibition zones (halos) around diffusion discs, which can be interpreted as changes in susceptibility to certain antibiotics (absolute values) (Appendix A). Figure 4 shows the relationship between changes in susceptibility within and between each group of samples. Due to the method used (diffusion discs), in the case of pathogens that were already extremally resistant to most of the antibiotics (e.g., resistant *P. aeruginosa*), the increase in resistance was practically impossible to observe (determining the minimal inhibitory concentration (MIC) would be required to assess these changes). Therefore, we would like to point out that no observed change in resistance does not eliminate the possibility of an increased level of resistance.

### 3.3. Genes Mediating Antibiotic Resistance

To establish whether mutagenic factors present during the flight affected genes mediating antibiotic resistance mechanisms *kpc* in CRE, *ndm*-1 in CRKP, *vim* in MDR *P. aeruginosa*, *van*B in VRE, and *mec*A in MRSA/VSSA and MRSA/VISA) we performed PCR analyses of those genes in isolates collected after the flight. None of the genes were lost in our samples.

## 4. Discussion

Experiments that use stratospheric balloons provide valuable data for several scientific areas. Alterations in bacterial metabolism have been demonstrated previously in bacteria isolated from patients and their response to stratospheric UV during a study focusing on viability and biochemical changes occurring after exposure to UVC radiation in the laboratory and stratosphere. Based on these findings, the authors have implied that Gram-positive bacteria are more resistant to UV radiation than Gram-negative bacteria [21]. We observed similar tendencies in our study.

Another field that can benefit from stratospheric flights is astrobiology. The National Aeronautics and Space Administration (NASA) has sent spore-forming Bacillus pumilus to the stratosphere in the E-MIST (Exposing Microorganisms in the Stratosphere) payload to examine whether bacteria carried on spacecraft sent to Mars can cause contamination. Mars’ harsh conditions are similar to those in the Earth’s stratosphere, which has been used to test bacterial viability in Mars-like conditions [22]. Similar studies can also focus on extremophilic bacteria—their astrobiological potential and whether extremophile-like organisms would be able to survive in Mars-like conditions [23].

The material of our study consisted of 11 strains classified into five species of human pathogens. These belonged to both Gram-positive and Gram-negative bacteria and were all non-spore-forming species. We noticed that 8 out of 11 strains had survived the harsh stratospheric conditions, but generally at a very low level. The highest survivability was observed in *Enterococcus* and *Staphylococcus* whereas the lowest was observed in *Pseudomonas* strains, both sensitive and resistant to antibiotics. Among Gram-negative bacilli, *Klebsiella pneumoniae* demonstrated the highest survivability. When considering the presented data, it is crucial to first identify to which physical factors each group of samples was exposed. The control group kept on the ground as pellets during the flight had the most favorable conditions. The possible survival limiting factors for this group include a lack of nutrients, desiccation, and accumulation of toxic metabolic products [24]. This can disrupt the functions of global regulatory systems and/or affect metabolism and the osmotic balance resulting in morphological changes or growth arrest [25]. The flight control group was covered with a layer of aluminum foil, which protected the samples from direct sunlight, UV, and some extent of space (ionizing) radiation and acted as thermal isolation [24,26]. However, those samples were exposed not only to the same conditions as the ground control group but additionally to ozone and hypobaric conditions [26]. Ozone is a well-known factor that causes DNA damage and oxidation of the components of cell membranes resulting in cell lysis. The antimicrobial activity of ozone is well documented in the literature and confirmed in practice. Ozone is widely used in the food industry for food preservation and in other decontamination technologies [27,28]. There is little research on the effects of hypobaric conditions on bacteria, but it seems that exposure to low pressure triggers changes in cell membrane polarization and the composition of fatty acids in cells. One would assume that hypobaric conditions would arrest the growth of bacteria, but there are no definitive data on that matter [29,30]. The experimental group experienced all the factors described above as well as those that the flight control samples were protected from [26]. During our study, samples sent to the stratosphere were exposed to doses of UV radiation 4 times higher than the usual exposure at sea level. The effects of UV radiation on living organisms depend on the wavelength and susceptibility of a particular species. Normally, UVC (200–280 nm) does not reach the ground because it is blocked by the ozone layer. Exposure to UVC is associated with the increased production of oxygen reactive species (ROS) leading to biomolecular damage as well as direct DNA damage, blocking DNA replication and RNA transcription [31]. Space (ionizing) radiation is mostly blocked by Earth’s geomagnetic field, but it increases with altitude. Although dosimeters implemented in the study did not reveal increased doses of ionizing radiation (perhaps due to the short duration of the flight and limited sensitivity of the dosimeters), we need to point out the detrimental effects of ionizing radiation on cells. Damage caused by ionizing radiation is widely associated with the production of reactive radicals leading to DNA damage by the modification of bases or direct strand breaks. It can result in cell death or favor mutational changes [32,33]. Fluctuations in temperature cause changes in cell structures, the composition and organization of cell membranes, and metabolism. Low temperatures affect fatty acids, decrease membrane fluidity, and cause conformational changes in DNA and proteins. Moreover, the function of several enzymes is impaired, which affects metabolic reactions and translation resulting in growth arrest [34].

After collecting the data and identifying the relevant factors for each group, we wanted to further examine the properties of the species used in our study to look for possible explanations for the results. One of the features that we have investigated is the variability of genomic content (GC) between bacterial species. The species in our experiment vary between those with high GC (~51% in *E. coli*, ~57% in *K. pneumoniae*, and ~67% in *P. aeruginosa)* and low GC content (~37% in *E. faecalis* and ~33% in *S. aureus)*. High GC is typically associated with species that live in a free environment prone to less-stable conditions [35]. However, studies also show that genomes rich in GC are more susceptible to damage caused by radiation [36]. That could be a possible explanation for why *E.coli* and both strains of *P. aeruginosa* in the experimental group (exposed to UV and ionizing radiation) did not survive the flight. Another important feature is the size of the genome. Radiobiology studies show that the bigger the genome, the more prone the cell is to DNA damage (especially double-strand breaks) [37]. The fact that *P. aeruginosa* has the largest genome (~6.8 Mbp) and that both of its strains exposed to the radiation did not survive the flight is in line with those findings. Moreover, some mechanisms of antibiotic resistance are associated with an increase in the size of the genome (e.g., plasmids, integrons, pathogenicity, and resistance islands), which corresponds to the fact that, in several cases, resistant strains (carrying various plasmids) potentially exposed to radiation showed less viability after the flight. Next, our study included Gram-negative and Gram-positive bacteria. The composition and organization of cell walls and other extracellular structures can also have implications on the resilience of the bacteria. Differences in this structure translate into different properties and mechanisms of the cell envelope stress response between Gram-positive and Gram-negative strains [38,39]. The structure of Gram-negative bacteria also makes them more prone to desiccation [38]. One should keep this in mind; however, species differ within the Gram-positive/negative groups, which can overshadow their general properties. The fact that survival rates of *K. pneumoniae* samples are so high compared to other Gram-negative species could be explained by the tendency of *K. pneumoniae* strains to produce thick and abundant polysaccharide capsules. These can potentially serve as storage for nutrients and water for the cell and provide additional mineral compounds enabling them to survive longer, even in their poor environment [40,41]. The combination of all these features may have contributed to the differences in the survival rates of bacteria during the stratospheric flight.

Most anti-drug resistance mechanisms (both mutational and plasmid-borne) are an additional energetic or metabolic expense for a cell [42]. Our results show that antibiotic susceptibility decreased quite consistently in the ground control group. When bacteria are exposed to mild forms of stress, an opportunity is provided for them to improve their ability to adapt and become resistant to subsequent, more extreme exposure through physiological adjustment, enabling reproduction [43]. This can be caused by single mutations or non-specific responses (metabolic changes or encapsulation) to stressors (desiccation and starvation) causing a decrease in drug activity or penetration inside the cell. In the experimental group (exposed to the harshest conditions), susceptibility tended to increase. Usually, the fitness costs of antibiotic resistance mechanisms are reduced by compensation mechanisms, most often by mutations. The other mechanisms underlying adaptive/directed mutation include stress-induced errors during DNA synthesis, the suppression of normal DNA repair checking and repair mechanisms, transient hyper-mutability, and stress-induced recombination processes. The environmental stress can also modulate plasmid numbers, reducing resistance due to fewer plasmid copy numbers or reducing gene amplification [44]. We speculate that cellular damage in this group was so significant that those mechanisms could no longer sustain homeostasis: The whole effort of the cell had to be focused on repair mechanisms. Therefore, antibiotic resistance, which is often associated with alternations in crucial cellular processes but is not necessary for survival (in the absence of antibiotics), had to be lost.

In the flight control group, bacteria experienced less damage than the experimental group, and compensating mechanisms were able to keep the cells alive and maintain some of their resistance mechanisms. Since none of the genes mediating antibiotic resistance mechanisms were lost during the flight, it further suggests that changes in antibiotic resistance had to occur on a post-transcriptomic level. Arguably, in cases of resistant strains, a decrease in susceptibility could not be observed due to limitations of the methodology, and therefore *K. pneumoniae* and *P. aeruginosa* (resistant variants) may also fall into this pattern. It is possible that *P. aeruginosa*, as potentially the most sensitive strain to the harsh environment, had to fight for survival already as part of the flight control group (shown as increased susceptibility) and the conditions experienced in the experimental group caused too much damage and death.

## 5. Conclusions

We were surprised by the results of the susceptibility testing. Several stressors likely had an impact on cellular homeostasis, and bacteria from our three groups had to develop different strategies to maintain balance.

We can conclude that most human non-spore-forming bacteria can survive harsh stratospheric conditions, promoting the loss of antibiotic resistance rather than favoring it.

## Figures and Tables

**Figure 1 ijerph-20-02787-f001:**
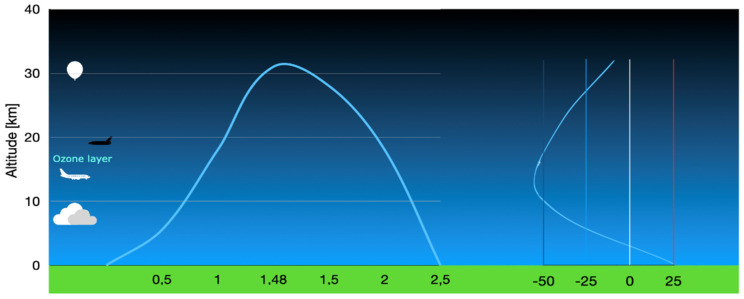
STRATOS mission flight profile. The flight was 2.5 h long, with approximately 1 h exposure to high levels of UVA, UVB, and UVC light. Balloon burst occurred in the stratosphere at approximately 31 km altitude. The graph on the right side visualizes temperature fluctuations.

**Figure 2 ijerph-20-02787-f002:**
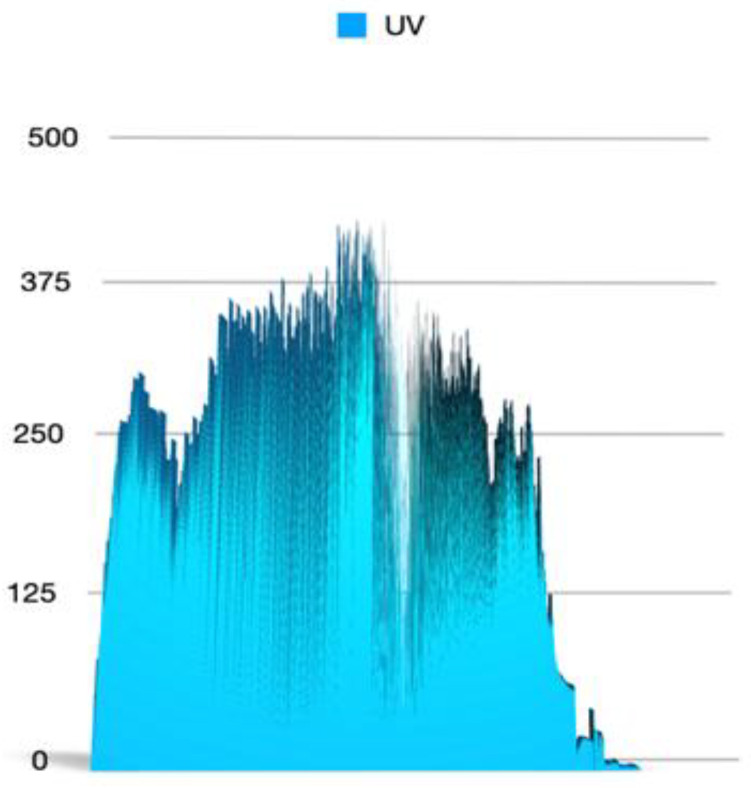
Data logs from the GUVA-S12SD 240–370 nm sensor reveal oscillations in UV light intensity values indicating rotational movement of the biological capsule. *Y*-axis shows activity sensor units in relation to the 2.5 h time of flight duration (*x*-axis).

**Figure 3 ijerph-20-02787-f003:**
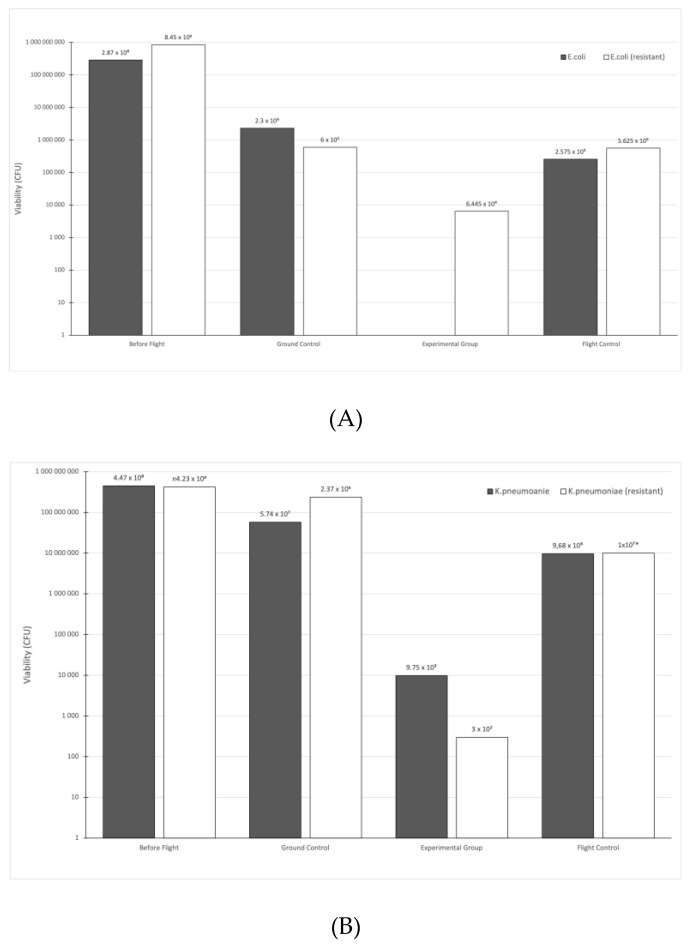
Bacterial viability (absolute values) of strains: *E. coli* (**A**), *K. pneumoniae* (**B**), *P. aeruginosa* (**C**), *E. faecalis* (**D**), and *S. aureus* (**E**) based on Colony-Forming Units (CFU). For each species, there is a model strain (susceptible to antibiotics) and a resistant strain (resistant to antibiotics). Because the amplitude of the numbers presented on the graphs is significant (low and high numbers), a logarithmic scale on the *y*-axis was used. A calculated CFU number is seen above each post.

**Figure 4 ijerph-20-02787-f004:**
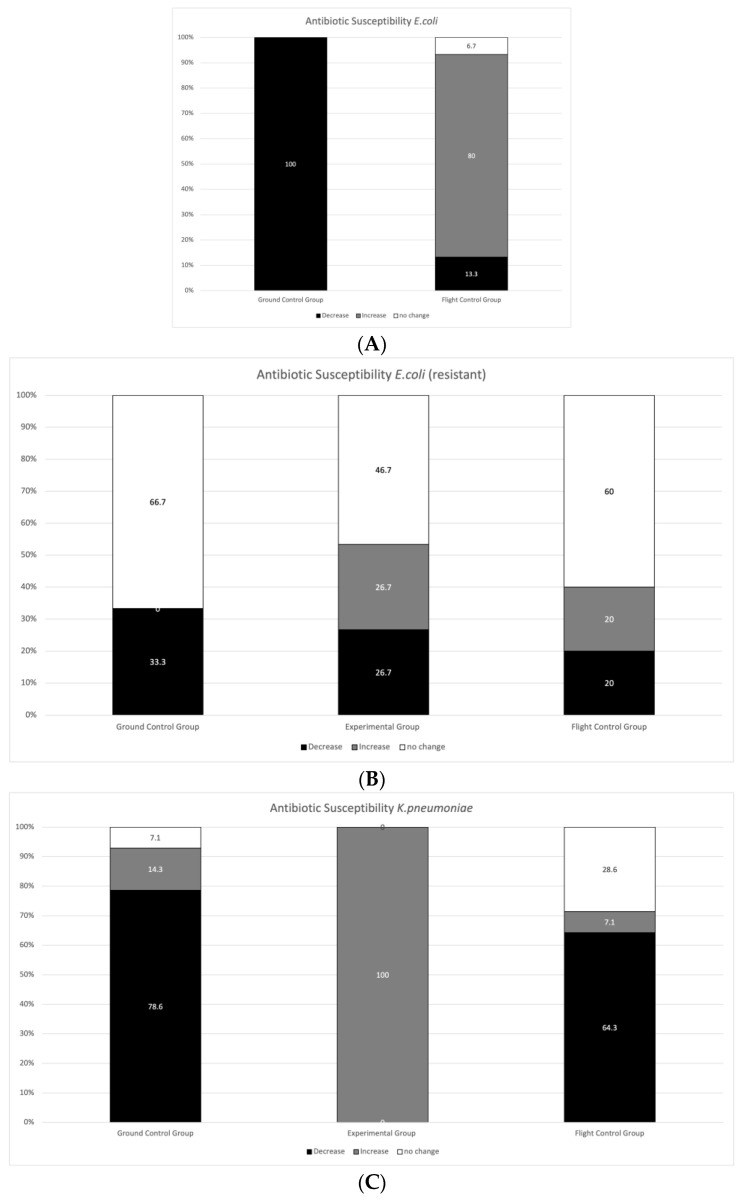
Relationship between changes in antibiotic susceptibility in each group. Antibiotic susceptibility of *E. coli* (**A**); Antibiotic susceptibility of resistant *E. coli* (**B**); Antibiotic susceptibility of *K. pneumoniae* (**C**); Antibiotic susceptibility of resistant *K. pneumoanie* (**D**); Antibiotic susceptibility of *P. aeruginosa* (**E**); Antibiotic susceptibility of resistant *P. aeruginosa* (**F**); Antibiotic susceptibility of *E. faecalis* (**G**); Antibiotic susceptibility of resistant *E. faecalis* (**H**); Antibiotic susceptibility of *S. aureus* (**I**);Antibiotic susceptibility of *S. aureus* (MRSA/VSSA) (**J**); Antibiotic susceptibility of *S. aureus* (MRSA/VISA) (**K**). Ground Control—samples kept on the ground; Experimental Group—samples flown to the stratosphere (without protection); Flight Control—samples flown to the stratosphere (with a protective layer of aluminum foil); Decrease—narrowing of the halo around diffusion discs/increase in MIC; Increase—widening of the halo around diffusion discs/decrease in MIC; No change—no change in the halo around diffusion discs/MIC. Experimental group is not indicated in cases where no growth was obtained. For absolute values of halos around diffusion discs see Appendix A.

**Table 1 ijerph-20-02787-t001:** Data logs collected at critical time points of the STRATOS mission flight revealing temperature extremes.

Time [min]	Temp [°C] 1	Temp [°C] 2	Temp [°C] 3	Temp [°C] 4
0	31.5	23.06	26.50	28.37
41	−45.94	−56.69	−54.88	−49.88
105	14.25	−15.88	−8.81	−7.69
133	−32.06	−51.81	−48.81	−46.06
156	25.56	24.25	24.50	24.75
206	24.94	22.69	23.75	25.37

**Table 2 ijerph-20-02787-t002:** Viability of the bacteria of each strain. Percentage values are normalized to the number of bacteria (Colony-Forming Units) measured before sample preparation (before the flight).

Strain	Viability (%)		
	Ground Control ^a^	Experimental Group ^b^	Flight Control ^c^
*E. coli*	8 × 10	0	8.97 × 10^−2^
*E. coli* (resistant)	7.1 × 10^−2^	7.6 × 10^−4^	6.7 × 10^−2^
*K. pneumoniae*	1.284 × 10	2.2 × 10^−3^	2.17
*K. pneumoniae* (resistant)	5.603 × 10	7.1 × 10^−5^	2.36 *
*P. aeruginosa*	1.7 × 10^−1^	0	3.4 × 10^−2^
*P. aeruginosa* (resistant)	2.8 × 10^−2^	0	7.6 × 10^−3^
*E. faecalis*	9.086 × 10	5.6 × 10^−4^	5.71 *
*E. faecalis* (resistant)	3.6 × 10	2.5 × 10^−4^	3.08 *
*S. aureus*	5.855 × 10	5.2 × 10^−4^	2.41
*S. aureus* (MRSA/VSSA)	4.989 × 10	4.4 × 10^−3^	2.11 *
*S. aureus* (MRSA/VISA)	3.268 × 10	2.6 × 10^−5^	5.26 *

Legend: ^a^ Ground Control—samples kept on the ground; ^b^ Experimental Group—samples flown to the stratosphere (without protection); ^c^ Flight Control—samples flown to the stratosphere (with a protective layer of aluminum foil); * Numbers are an estimation due to the excessive growth, which made counting the colonies impossible.

## Data Availability

Not applicable.

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
