# Peer review of "The Impact of Harsh Stratospheric Conditions on Survival and Antibiotic Resistance Profile of Non-Spore Forming Multidrug Resistant Human Pathogenic Bacteria Causing Hospital-Associated Infections"

_ijerph, 2023, doi:10.3390/ijerph20042787_

Round 1
Reviewer 1 Report
This manuscript describes an experiment in which bacteria are exposed to a flight to the upper atmosphere. The authors observed which bacterial species survived, as well as changes in their antibiotic resistance compared with control samples. I feel that this research would be of interest to the readers of this journal, however, some issues should be addressed.
While the long general introduction on antibiotic resistance in bacteria is appreciated, most of the material is not directly relevant to the current work, and should therefore be saved for a different venue. Instead, more references that specifically relate with stress-induced changes in bacteria should be included.
The data in figure 4 is presented in an unnecessarily confusing way. Showing the raw data showing the halos around the diffusion discs would be more straightforward.
(Line 493) "In the experimental group (exposed to the harshest conditions) susceptibility tended to increase...We speculate that cellular damage in this group was so significant that those mechanisms...the whole effort of the cell had to be focused on repair mechanisms."
I feel this is one of the major findings of the work, and the authors should elaborate on this topic. As noted above, more citations in the literature should be included on the effects of hazardous environments on antibiotic resistance.
Some typos exist, such as "O3" in line 116, which should be "O3".
Author Response
Dear Reviewer 1,
on behalf of the other authors and my own I would like to thank You for the reviewing and carefully evaluating of our manuscript. Referring to the suggestions and comments contained in the review, I would like to give You a response to, as follow below.
- We re-wrote the “Introduction” text to make it more coherent.
- Regarding to comment to Figure 4: We put the raw data showing the halos around the diffusion discs in “Supplementary_Materials” as Table S4.
- We elaborated and improved the “Discussion” section. Additional citations were added and the references were also re-matched.
- The ‘O3’ was corrected to ‘O3’.
- A minor correction of the English language was also provided to improve the quality the of the manuscript.
All changes have been marked and highlighted for clarity and easy viewing. We submit our revised manuscript with tracked changes to facilitate the review in the file called “ijerph-2155811_Revision_with_marked_changes”.
I kindly request You to re-evaluate the revised manuscript.
Sincerely,
Ksenia Szymanek-Majchrzak
Reviewer 2 Report
The manuscript of Górecki and co-authors (The impact of harsh stratospheric conditions on survival and antibiotic resistance profile of non-spore forming multidrug resistant human pathogenic bacteria causing hospital-associated infections) deals with the evaluation of different bacteria (antibiotic- resistant and non-resistant) under the conditions of a balloon flight into the stratosphere.
Such an investigation is useful and also provides detailed insights into the problem of multi-resistant strains.However, there are some inconsistencies in this study.
Point 1: lines 153-157: please explain the specific conditions. Were the pellets frozen directly without any protection to -70°C, like the addition of glycerol? This could explain some of the dramatic loss of survivors Point 2: Table 2 is confusing. Here the variability is given in %? To my knowledge the ground control than should be 100 and the rest less or 0. So what is the meaning of 6.97 x 10-2, for example? Pont 3: Figure 3: how many samples were analysed for each set up? In case there was more than 1 samples, please indicate the arrow bars. Point 4: Figure 4: (A) why is the sample E. coli not resistant indicated for the experimental group, when there are no survivors as indicated in Figure 3? In case of samples where the MIC could not estimated, please indicate in the figure.
Author Response
Dear Reviewer 2,
on behalf of the other authors and my own I would like to thank You for the reviewing and carefully evaluating of our manuscript. Referring to the suggestions and comments contained in the review, I would like to give You a response to, as follow below.
- We did not use additional protection (e.g. glycerol), because we wanted to provide conditions as close as possible to those to which bacteria are exposed in nature (they reach the stratosphere in the form of microaerosol or on micropollen).
- Percentage values are normalised to the number of bacteria (Colony Forming Units) measured before sample preparation (before the flight, 100%).
- Data refer to single samples.
- Figure 4 was improve as suggested. In addition, referring to Table S4 in the Supplementary_Materials will be helpful in reading and interpreting the results of Figure 4.
All changes have been marked and highlighted for clarity and easy viewing. We submit our revised manuscript with tracked changes to facilitate the review in the file called “ijerph-2155811_Revision_with_marked_changes”.
I kindly request You to re-evaluate the revised manuscript.
Sincerely,
Ksenia Szymanek-Majchrzak